# Classification of Dental Radiographs Using Deep Learning

**DOI:** 10.3390/jcm10071496

**Published:** 2021-04-03

**Authors:** Jose E. Cejudo, Akhilanand Chaurasia, Ben Feldberg, Joachim Krois, Falk Schwendicke

**Affiliations:** 1Department of Oral Diagnostics, Digital Health and Health Services Research, Charité–Universitätsmedizin Berlin, 14197 Berlin, Germany; jose-eduardo.cejudo@charite.de (J.E.C.); ben.feldberg@charite.de (B.F.); Joachim.krois@charite.de (J.K.); 2ITU/WHO Focus Group AI on Health, Topic Group Dentistry, 1211 Geneva, Switzerland; chaurasiaakhilanand49@gmail.com; 3Department of Oral Medicine and Radiology, King George’s Medical University, Lucknow 226003, Uttar Pradesh, India

**Keywords:** artificial intelligence, classification deep learning, dental, machine learning, radiographs, teeth

## Abstract

Objectives: To retrospectively assess radiographic data and to prospectively classify radiographs (namely, panoramic, bitewing, periapical, and cephalometric images), we compared three deep learning architectures for their classification performance. Methods: Our dataset consisted of 31,288 panoramic, 43,598 periapical, 14,326 bitewing, and 1176 cephalometric radiographs from two centers (Berlin/Germany; Lucknow/India). For a subset of images L (32,381 images), image classifications were available and manually validated by an expert. The remaining subset of images U was iteratively annotated using active learning, with ResNet-34 being trained on *L*, least confidence informative sampling being performed on U, and the most uncertain image classifications from U being reviewed by a human expert and iteratively used for re-training. We then employed a baseline convolutional neural networks (CNN), a residual network (another ResNet-34, pretrained on ImageNet), and a capsule network (CapsNet) for classification. Early stopping was used to prevent overfitting. Evaluation of the model performances followed stratified k-fold cross-validation. Gradient-weighted Class Activation Mapping (Grad-CAM) was used to provide visualizations of the weighted activations maps. Results: All three models showed high accuracy (>98%) with significantly higher accuracy, F1-score, precision, and sensitivity of ResNet than baseline CNN and CapsNet (*p* < 0.05). Specificity was not significantly different. ResNet achieved the best performance at small variance and fastest convergence. Misclassification was most common between bitewings and periapicals. For bitewings, model activation was most notable in the inter-arch space for periapicals interdentally, for panoramics on bony structures of maxilla and mandible, and for cephalometrics on the viscerocranium. Conclusions: Regardless of the models, high classification accuracies were achieved. Image features considered for classification were consistent with expert reasoning.

## 1. Introduction

Radiographic images are ubiquitous in many medical settings; especially in dentistry, imaging is at the cornerstone of many patients’ dental voyage, from diagnosis, to treatment planning, to conducting and re-evaluating therapies. Dental images involve photographs, radiographs, 3-D scanning, cone beam computed tomography (CBCT), video data, etc.; no discipline in medicine takes more radiographs than dentistry [1].

In today’s digital practice infrastructure, such imagery should be indexed and digitally stored in archiving or patient management systems, allowing one to retrieve these data easily for diagnostics, treatment, and monitoring. Traditionally, this indexing and storing process has been done manually.

Image indexing, i.e., assigning a label to an image (like “periapical radiograph of tooth 47”), is considered an image classification task. Such classification tasks can be automated using deep learning, which is a branch of machine learning that excels on high-dimensional data such as text and images [2]. Convolutional neural networks (CNNs), one of the most common deep learning architectures, have been employed in dentistry to detect and classify image objects like teeth or restorations as well as to detect, classify, and segment pathologies like dental caries or apical lesions on dental radiographs [3]. Software products building on such deep CNNs are supposed to assist the dental practitioner in image analysis, including speeding up the diagnostic process, improving accuracy, and easing comprehensive reporting [4].

To retrospectively assess the available large-scale image pool in most dental settings as well as to avoid manual labelling of prospective imagery, deep learning can be employed, too. Automated classification of dental image types, like different radiographs (e.g., panoramics, bitewings, periapicals, and cephalometrics), would allow one to make better use of the vast existing data in dental practices and hospitals as well as easing the dental workflow for new imagery. For example, such automated classification could be part of existing software suites, e.g., radiographic viewing software or patient management software, allowing automated labeling and appropriate storage and indexing of new imagery. Additionally, such classification could be used for research purposes to mine existing unstructured image databases, thus enabling better usage of data for scientific objectives.

We aimed to train and test deep learning models for classifying dental radiographs, comparing three popular network architectures for their classification performance. Moreover, we applied elements of explainable artificial intelligence (XAI) to visualize salient areas most relevant for the classification, helping to understand and interpret the models’ output.

## 2. Methods

### 2.1. Study Design

In the present study, three deep learning model architectures for classification of dental radiographs were trained, validated, and tested in a supervised learning setting, i.e., employing a dataset with pairs of images and labels. The dataset was composed of four types of radiographic images, i.e., panoramics, bitewings, periapicals, and cephalometrics. A combination of active learning (AL) as part of the annotation pipeline, a weighted loss function for handling the imbalance in the classes, early stopping to prevent overfitting, and transfer learning were used. Ten-fold cross validation scheme was applied to account for uncertainty. We further visualized the salient areas of the images that had most importance in the models’ output, i.e., by leveraging XAI. Reporting of this study follows the Standards for Reporting of Diagnostic Accuracy Studies (STARD) guidelines [5], the Checklist for Artificial Intelligence in Medical Imaging (CLAIM) [6], and the Checklist for Artificial Intelligence in Dental Research [7].

### 2.2. Data and Sampling

The dataset consisted of a total of 90,388 radiographs originating from routine care provided at Charité–Universitätsmedizin Berlin, Germany (Charité) and King George’s Medical University (KGMU), Lucknow, India. Data collection was ethically approved (Charité ethics committee EA4/080/18). Charité provided in total 31,288 panoramic, 43,598 periapical, and 14,326 bitewing radiographs, whereas KGMU provided 1176 cephalometric radiographs (Table 1). The data were collected between 2016 to 2018 and 2011 to 2019 at KGMU and Charité, respectively. The patients’ ethnicity at KGMU was Indian, while further meta-data were not available. The patients’ ethnicity at Charité can be assumed to be predominantly Caucasian, while it is worth noting that Berlin hosts a large Turkish and Arabic community, too (totaling approx. 9% of the city’s total population). As mentioned, age and gender information was available from Charité only. There were 49.1% male and 47.3% female patients in the dataset (the remaining portion was not reported). The mean (SD, min–max) age was 47.4 (20.4, 3–99) years. The data from both centers were generated using radiographic machines from the manufacturer Dentsply Sirona (Bensheim, Germany), mainly Orthophos SL, Orthophos XG3D, and XIOS Plus (in toto 66.3%); machines from Dürr Dental (Bietigheim-Bissingen, Germany), mainly Vista Scan (22.6%); and machines from Carestream (Rochester, New York, USA), mainly CS 9300 (1.8%). For approx. 10% of the radiographs, the manufacturer was not reported. All image data were extracted from PACS as DICOM files, pseudo-anonymized and exported as JPEG images; no further image preprocessing was applied.

For a subset of images L (32,381 images), information of its category (panoramic, bitewing, periapical, or cephalometric) were available. Notably, the proportion of images per category varied greatly, with the dataset being heavily imbalanced (panoramic, 46.38%; bitewing, 22.23%; periapical 27.74%; and cephalometric 3.63%). For another subset of images U (58,007 images), no such information was available.

### 2.3. Reference Test; Active Learning

The subset L was manually reviewed by an experienced dentist (BF), and its classification information was validated. The remaining subset of images U was iteratively annotated using active learning (AL) [8]. AL is a branch of machine learning that aims to minimize the annotation effort of human annotators. A model (we used ResNet-34; model details are described below) was trained on the labeled data L. In order to query data points from U for labeling by a human annotator, the least confidence informative sampling approach was adopted [9]. Such sampling uses a measure of the informativeness (here the confidence of the model inference was used) for deciding which queries are provided to the annotator for labeling. By doing so in each iteration, the annotator was provided with the most uncertain image instances. Once reviewed and labeled by the human expert, the existing labeled data were augmented with the newly annotated data at each iteration. Hence, the amount of labeled data increased with every iteration, as did the performance of the model. In total, 10 iterations were needed to label the complete dataset U. This approach turned out to be faster and less work-intensive for the annotators than labeling the entire unlabeled dataset from scratch. Notably, the ResNet-34 model trained for the AL task was not used for the experiments outlined below.

### 2.4. Sample Size

As a comprehensive sample of available and labelled imagery was used, no formal sample size estimation was performed.

### 2.5. Model, Model Parameters, and Training

Numerous neural network architectures for computer vision deep learning models have been proposed [10]. CNNs leverage the concept of convolution layers [2,11], which serve as an effective feature extractor for images. More recently, residual networks were introduced, which add skip connections between layers and have demonstrated high performance on computer vision tasks [10]. Capsule networks [12] have also drawn attention given their novel architecture and routing mechanism and have achieved high performances in medical image analysis [13]. In the present study, the performance of a baseline CNN, a residual network (ResNet), and a capsule network (CapsNet) were evaluated for radiographic image type classification.

The baseline CNN was composed of two convolutional blocks with 16 and 32 kernels, a ReLU activation function, and a Max-Pooling layer. Its classification head was composed of two fully connected layers of 512 units followed by a ReLU and an output layer with a softmax activation function. The weights of the baseline CNN were initialized randomly.

The residual network was a ResNet-34 model architecture, pretrained on the ImageNet dataset. The classification head was replaced by a fully connected layer of outputs equal to the number of labels in our dataset, i.e., four. A softmax activation function was applied to the final layer. For both the baseline CNN and ResNet-34, weighted cross entropy was used as the loss function. The weights were inversely proportional to the fraction of each category to account for the class imbalance. The misclassification of categories with fewer instances resulted in a larger loss, forcing the model to learn to correctly classify the minority classes.

For the capsule network, a feature extraction module of CapsNet with a single convolutional layer of 256 kernels followed by a ReLU activation was used. The features were fed to a layer of 32 capsules followed by the output layer with 4 capsules, all of them of 16 dimensions. The margin loss was used with the original parameters [12], and the network was randomly initialized.

Prior to feeding data to the models, the images were reshaped to arrays of 224, 224, and 3 for ResNet and the baseline CNN. Lower resolutions (64, 64, and 3) were used for the capsule network due to computational constraints. For ResNet, images were normalized with the mean and standard deviation of the ImageNet dataset. No image augmentation was used in the experiments as maximizing the generalizability of the models was not the focus of the study. The batch size was set to 16, and the Adam optimizer with learning rate 0.0001 was used for training the models. Early stopping [14] was used to prevent overfitting, by monitoring the validation loss and stopping training when the loss did not decrease after five epochs. In all cases, the input of the models was an RGB image obtained by triplicating the single channel raw image, and the output was a distribution over the different image classes (panoramic, bitewing, periapical, or cephalometric). The resulting output class was the one with the highest score. The models were trained for 60 epochs on two NVIDIA Quadro RTX 6000 graphic cards. Modelling was performed using the PyTorch implementation of torchvision.

### 2.6. Evaluation, Uncertainty, and Explainability

Classical multiclass classification metrics were used for measuring the performance of the models on the test set, namely, accuracy, precision, sensitivity, F1-score, and specificity. Further, the receiver operating characteristic (ROC) curve and the confusion matrix were computed. Evaluation of the model performances followed a stratified k-fold cross-validation [15] with 10 train, validation, and test splits (73,214; 8135; and 9039 images, respectively), thereby accounting for the original distribution of data across the splits. One of the main challenges of deep learning applications is the difficulty to interpret the decisions made by the deep learning systems. Therefore, Gradient-weighted Class Activation Mapping (Grad-CAM) [16] was used, which provides visualizations of the weighted activations maps, resulting in visualizations of the salient regions of an image that are relevant for the classification outcome, to overcome this challenge.

## 3. Results

The three different models all showed high accuracy (Table 2), with significantly higher accuracy, F1-score, precision, and sensitivity of ResNet over the baseline CNN and the CapsNet model (*p* < 0.05). The specificity was not significantly different between the three models. Misclassification was most common for bitewings being classified as periapicals.

If inspecting the training history (Figure 1), all performance metrics increased over the number of epochs until converging, with ResNet achieving the best performance at small variance and fastest convergence. Performance differences are further reflected in Figure 2, showing the ROC curves for the different models and the different classes. For the baseline CNN and CapsNet, the accuracy for periapicals was somewhat lower than for other imagery, while ResNet did not show such differences. Our findings are further reflected by the confusion matrix (Figure 3).

Assessing the salient features of the best performing model (ResNet; Figure 4) indicated that for bitewings the model had the highest activation in the space between the teeth but not the roots, while for periapicals the model focused on the interdental space. For panoramics, bony structures of the maxilla and the mandible were most relevant, while for cephalometrics the viscerocranium (not the neurocranium) was most relevant.

## 4. Discussion

In the present study, we developed a classification model for dental radiographs. Such model itself does not add diagnostic value, as obviously classification of radiographs is a task that is easy to perform for dentists. However, it relieves the human expert from this task in two exemplary scenarios: retrospectively classifying existing dental radiographs and prospectively labelling new radiographic imagery, which could be useful when radiographs are automatically passed on from radiographic software or image archives, allowing the dentist to omit a manual classification step and thereby easing his workflow. We further benchmarked three different model architectures for this purpose, finding that all three showed high accuracy, but that ResNet-34, pretrained on ImageNet, outperformed the other architectures. We further showed that the most notable misclassifications occurred between bitewing and periapicals. Using elements of explainable AI, we showed that the models considered characteristic features like the interarch space on bitewings, interdental areas on periapicals, bony structures on panoramics and the lateral view of the viscerocranium for cephalometrics to come to a classification decision. Given that image classification is a complex task, with deep learning being widely considered as black box, and considering that significant bias has been identified in classification tasks [17], our XAI results enhance the confidence into the classification models.

Our findings require some more detailed discussion. First, and as laid out, the developed models will not increase a dentist’s accuracy for pathology detection but will ease the clinical workflow, allowing automated classification of any image passed onto AI-based software systems from existing practice software suits. It will further allow one to automatically mine existing databases, e.g., from clinics or hospitals, for both research and business intelligence purposes.

Secondly, benchmarking of the network architectures has not been widely performed in dentistry. We selected the three architectures as described, for experimentation purposes, being mainly interested in trying different types of models for solving this particular problem. We obtained similar levels of performance for the different architectures, confirming that regardless of the specific architecture, high accuracies are possible given sufficient data being available.

Thirdly, we need to highlight that only for ResNet, transfer learning was applied, eventually outperforming both the CNN and the capsule network. We attribute this gap in performance to the initialization of ResNet with pre-trained weights rather than to architectural factors. This confirms the importance of pretraining; the pretrained model has been exposed to a larger amount of data, has learned rich features, and leverages this learning for the finetuning, often resulting in faster convergence and better performance. ResNet already had accuracy >97% when trained for only one epoch on the validation test. The other two models, which were initialized randomly, exhibited a lower convergence rate, and their standard deviation was higher.

Fourthly, active learning proved to be an efficient method to get large amounts of images annotated swiftly. In our case we already had one third of the total amount of images labeled, which allowed us to start with a model with high performance. The annotators found correcting batches of predictions of this model less work-intensive than reviewing a monolithic unlabeled dataset. Future studies should evaluate if such strategy is also useful for other tasks, e.g., segmentation of pathologies.

Last, classification accuracy was generally high; the most notable misclassification was between periapicals and bitewings. This might be because we drew a comprehensive sample of bitewings, without any kind of exclusion criteria applied towards image quality or positioning. In a number of bitewings, positioning was suboptimal and the image mostly depicted only one dental arch; such image may be considered as periapical of, for example, the molar region and is hard to discriminate from “true” periapicals. We decided to allow for imagery of varying quality and positioning, as such artefacts will occur in practice, too, and a useful model should be robust in more challenging datasets as well.

This study comes with a number of strengths and limitations. First, it was built on one of the largest datasets of dental radiographs employed for deep learning so far; this allowed to reach excellent accuracies and likely increased the robustness of our findings. Second, to our knowledge this is the first study aiming to classify four different dental radiograph types, a task which—as laid out—is clinically useful (while not increasing accuracy, it eases the dental workflow). Third, using elements of XAI we were able to assess the reasoning behind classification decisions, scrutinizing them for any bias and thereby increasing trust and confidence. Fourth, and as a limitation, the differences in pretraining strategies may have biased our findings, as discussed, which is why the performed benchmarking should be interpreted with caution. Last, the developed models were trained and tested from various image subtypes stemming from two different centers, while each subtype data stemmed from one single center only. We hence cannot infer any cross-center generalizability, a property recently found highly relevant for computer vision tasks on dental radiographs [18]. Future studies should aim to test the developed models on more heterogeneous data.

## 5. Conclusions

Within the present study, we compared three different deep learning architectures (ResNet, CapsNet, and Baseline CNN) to classify dental radiographs. Regardless of the models, high classification accuracies were achieved, while a pretrained ResNet performed best. Misclassification was most common between bitewings and periapicals. The image features considered for classification were consistent with expert reasoning.

## Figures and Tables

**Figure 1 jcm-10-01496-f001:**
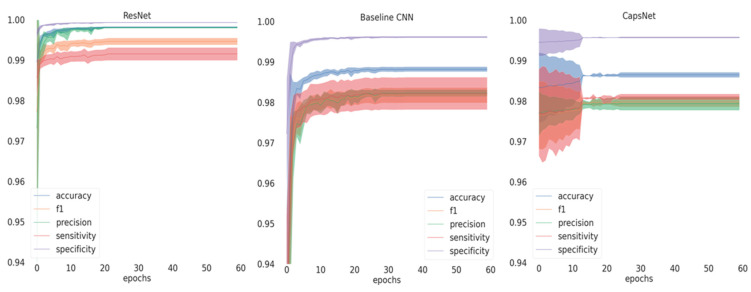
Training history depicting the mean and standard deviation of the metrics on the validation set across splits for the three models. The solid lines represent the mean and the shaded area the standard deviation. CNN: Convolutional Neural Network.

**Figure 2 jcm-10-01496-f002:**
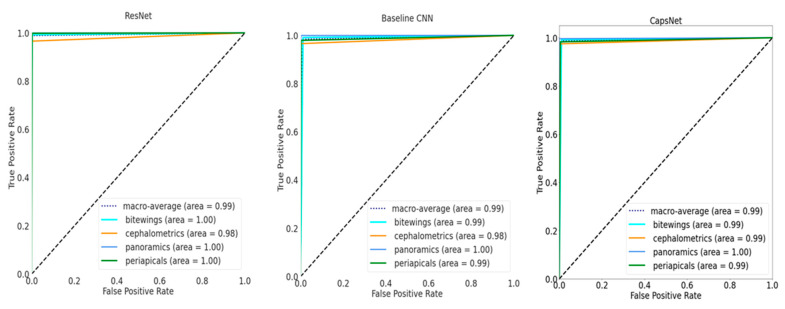
Receiver operating characteristic (ROC) curves on the test set for the three models. The area-under-the-curve for each class are shown. In addition, the macro-average across all classes is shown.

**Figure 3 jcm-10-01496-f003:**
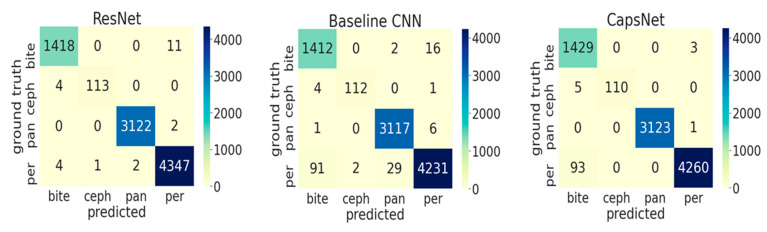
Confusion matrix on test data (9039 images) for the three models (ResNet, Baseline CNN, and CapsNet). The diagonal elements correspond to the correctly classified images, and the off-diagonal elements reflect the misclassifications. Abbreviations: per: periapicals; pan: panoramics; ceph: cephalometrics; and bite: bitewings.

**Figure 4 jcm-10-01496-f004:**
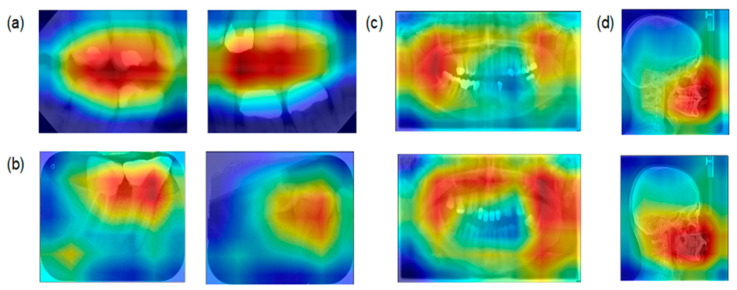
ResNet Gradient-weighted Class Activation Mapping (Grad-CAM) activation maps for (**a**) bitewings, (**b**) periapicals, (**c**) panoramics, and (**d**) cephalometrics. Red areas indicate the most salient regions for the classification, and blue areas the less important regions.

**Table 1 jcm-10-01496-t001:** Characteristics of the image dataset.

Image Type	Data Provider	Number of Images (%)	Mean Age (SD, min–max)	Gender (%)
Overall	---	90,388 (100)	47.6 (20.2, 14–96)	M: 50.2, F 49.1
Panoramics	Charité	31,288 (34.6)	46.3 (22.8, 14–96)	M: 50.0, F 49.9
Bitewings	Charité	14,326 (15.8)	37.0 (15.0, 14–89)	M: 50.8, F 48.1
Periapicals	Charité	43,598 (48.2)	53.4 (17.7, 14–94)	M: 51.2, F 47.9
Cephalometrics	KGMU	1176 (1.3)	---	---

KGMU: King George’s Medical University, SD: Standard Deviation.

**Table 2 jcm-10-01496-t002:** Performance of the different models for dental radiograph classification (mean, 95% CI). Statistical significance between models is indicated in bold (*p* < 0.05).

Model	Parameters	Accuracy	F1-Score	Precision	Sensitivity	Specificity
ResNet	21 M	0.997 (0.996, 0.998)	0.991 (0.989, 0.993)	0.996 (0.995, 0.997)	0.987 (0.983, 0.991)	0.999 (0.998, 0.999)
Baseline CNN	46 M	0.987 (0.986, 0.988)	0.980 (0.979, 0.982)	0.982 (0.980, 0.984)	0.979 (0.976, 0.982)	0.996 (0.995, 0.996)
CapsNet	17 M	0.989 (0.988, 0.990)	0.984 (0.981, 0.986)	0.985 (0.983, 0.986)	0.983 (0.980, 0.986)	0.997 (0.996, 0.997)

## Data Availability

Data cannot be made available given data privacy reasons.

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
