# Peer review of "Classification of Dental Radiographs Using Deep Learning"

_jcm, 2021, doi:10.3390/jcm10071496_

Round 1

Reviewer 1 Report

Thank you for submitting the paper "Classification of Dental Radiographs Using Deep Learning". A very well organized and quite striking investigation.

Just a few comments:

Reference 10: the pages are missing, 

Reference 18 is bad, why does it put accepted at the end?

What is the importance of this study? Do you think it has any useful application for our day to day as dentists?

Author Response

Reference 10: the pages are missing --> Corrected.

Reference 18 is bad, why does it put accepted at the end? --> Corrected.

What is the importance of this study? Do you think it has any useful application for our day to day as dentists? --> This was added.

Reviewer 2 Report

It's a very interesting topic. However, there is not enough explanation for how differentiating radiographs can benefit the clinic. Although it is necessary to use artificial intelligence to diagnose or specific disease performed in previous studies, how can the classification of simple radiographs in the present study be clinically helpful? Was it simply done to classify data for future research? The clinical necessity and further studies in the future need to be fully explained in the introduction and discussion sections. It's a very interesting topic, so if the introduction and discussion sections are reinforced, I recommend posting.

Author Response

We expanded on the rationale and usefulness of this study.

Reviewer 3 Report

In my opinion, this work is not suitable for publication in this journal.

The authors do not adequately justify their work and do not describe the clinical implication. The authors do not justify the novelty of their results.

Four of the eighteen bibliographical references used are by the authors of the manuscript. This number of bibliographical references can be considered excessive.

Author Response

The authors do not adequately justify their work and do not describe the clinical implication. The authors do not justify the novelty of their results.

--> This is the first study of its kind, so novelty is given, we nevertheless expand  more on its relevance and the clinical usefulness.

Four of the eighteen bibliographical references used are by the authors of the manuscript. This number of bibliographical references can be considered excessive.

--> We cite the only extensive scoping review, a checklist, the only study only generalisability in the field and a narrative critical review focusing in the limitations of AI in dentistry. We do not see this as excessive but justified.

Round 2

Reviewer 2 Report

The queries and comments have been responded to as suggested.

Reviewer 3 Report

After revision of the manuscript the authors have not improved the work.